# Gradient Information for Representation and Modeling

**Jie Ding**

School of Statistics

University of Minnesota

Minneapolis, MN 55455

dingj@umn.edu

**Robert Calderbank**

Department of Electrical and Computer Engineering

Duke University

Durham, NC 27708

robert.calderbank@duke.edu

**Vahid Tarokh**

Department of Electrical and Computer Engineering

Duke University

Durham, NC 27708

vahid.tarokh@duke.edu

## Abstract

Motivated by Fisher divergence, in this paper we present a new set of information quantities which we refer to as gradient information. These measures serve as surrogates for classical information measures such as those based on logarithmic loss, Kullback-Leibler divergence, directed Shannon information, etc. in many data-processing scenarios of interest, and often provide significant computational advantage, improved stability, and robustness. As an example, we apply these measures to the Chow-Liu tree algorithm, and demonstrate remarkable performance and significant computational reduction using both synthetic and real data.

## 1  Introduction

A standard step in data fitting and statistical model selection is the application of a loss function (sometimes referred to as scoring function) of the form $s : (y, p) \mapsto s(y, p)$, where $y$ is the observed data and $p(\cdot)$ is a density function. In this context, it is assumed that the smaller $s(y, p)$, the better $y$ fits $p$. A class of such functions that exhibit desirable statistical properties has been studied in the context of proper scoring functions [1]. As a special case, the logarithmic loss $s_L(y, p) = -\log p(y)$ has served as the cornerstone of classical statistical analysis because of the intimate relation between logarithmic loss and the Kullback-Leibler (KL) divergence. In fact, the KL divergence from a density function $p$ to the true data-generating density function $p_*$ can be written as $\mathbb{E}\{s_L(y, p)\} + c$, where the expectation of $s_L(y, p)$ is taken under $p_*$), and $c$ is a constant that only depends on $p_*$. Therefore, minimizing the sample average $n^{-1} \sum_{i=1}^{n} s_L(y_i, p)$ over a set of candidates density functions $p$ asymptotically amounts to finding the closest candidate $\hat{p}$ to $p_*$ in KL divergence.

A notable use of the logarithmic loss function is in maximum likelihood estimation for parametric models. By minimizing $n^{-1} \sum_{i=1}^{n} s_L(y_i, p_\theta)$ over $\theta \in \Theta$ for some parameter space $\Theta$, an estimate $\hat{\theta}$ is obtained to represent the data generating model. Some commonly used objective loss function such as cross-entropy loss for classification and squared loss for regression can be regarded as special cases of the logarithmic loss function. Another important use of the logarithmic loss is in model comparison

and model selection. In the presence of multiple candidate models, data analysts have to follow a model selection principle to select the most appropriate model for interpretation or prediction. The log-likelihood function, which can be regarded as logarithmic loss evaluated at observed data, play a crucial role in most state-of-the-art principles, including information criteria, Bayesian likelihood, Bayes factors (see, e.g., [2, 3] and the references therein). The logarithmic loss and KL divergence are also foundational in inference and information processing, exemplified by their use in variational inference [4], contrastive divergence learning [5], learning with information gains [6–8], etc.

Is the logarithmic loss always the best choice? In a series of recent works, a new loss function (also referred to as the Hyvarinen scoring function) [9] has been proposed as a surrogate for logarithmic loss function for statistical inference and machine learning. It is defined by

$$s_H(y, p) = \frac{1}{2} \|\nabla_y \log p(y)\|^2 + \Delta_y \log p(y), \tag{1}$$

where $\nabla$ denotes the gradient and $\Delta$ denotes the Laplacian, and $p$ is defined over an unbounded domain. It was first proposed in the context of parameter inference, which can produce (in a way similar to the logarithmic loss) a consistent estimation of $\theta$ of a probability density function $p_\theta(y)$ [9]. It was shown that $s_H(y, p)$ is computationally simpler to calculate in most cases, particularly in the case of intractable normalizing constants. This enables a richer class of models for prediction given the same amount of computational resources. It was discovered that the Hyvarinen loss also enjoys desirable properties in Bayesian model comparison and provides better interpretability for Bayesian model selection in the presence of vague or improper priors, compared with classical Bayesian principles based on marginal likelihoods or Bayes factors [10, 11].

On the other hand, from an information theoretic view, the differential entropy (for a random variable) is the expectation of its logarithmic loss, the mutual information (between two random variables) is the KL divergence from the product density function to the joint density function, and the mutual information is linked to differential entropy through the chain rule. This view is crucial to motivate our new information measures. Motivated by the definition of Shannon's differential entropy, it is natural to define the "entropy" for the Hyvarinen loss. This turns out to be intimately related to Fisher divergence. In fact the classical mutual information may be re-defined based on Fisher divergence instead of KL divergence. It turns out that these quantities still can be interpreted in information theoretic manner.

The main contributions of our work are described next. First, motivated by some recent advances in scoring functions, we propose a set of information quantities to measure the uncertainty, dependence, and stability of random variables. We study their theoretical properties, resemblance and difference to the existing counterpart measures widely used in machine learning. Second, we provide interpretations and applications of the proposed gradient information, e.g. to fast tree approximation and community discovery. Third, we point out some interesting directions enabled by gradient information, including a new form of causality for predictive modeling, channel coding where the stability of channel capacity is of a major concern, and general inequalities that could have been highly nontrivial from an algebraic point of view.

The paper is outlined as follows. In Section 2, we introduce the Hyvarinen loss function and extend its scope from unbounded to bounded continuous random variables. We introduce a set of quantities in Section 2.3 referred to as gradient information measures, and study their properties, interpretations, and implications for machine learning. In Section 3, we provide some applications of the proposed concepts, including a new algorithm for graphical modeling that parallels the classical Chow-Liu tree algorithm (but from an alternative perspective that can enjoy computational benefit), a new supervised learning algorithm, and a community discovery algorithm. Proofs and three additional applications of gradient information are included in the supplementary material.

## 2 Gradient Information and Its Properties

### 2.1 Fisher divergence and Hyvarinen loss

We focus on multidimensional continuous random variables (often denoted by $Y \in \mathbb{R}^d$) in this paper unless otherwise stated. We use $p$ and $\mathbb{E}$ to denote the density function and expectation with respect to the distribution of $Y$, respectively. The $j$th entry of $Y$ is denoted by $Y_j$ ($j = 1, \ldots, d$). Let $[Y, Z]$ denote the joint vector that consists of $Y$ and $Z$. We often use upper and a lower case letters to respectively denote a random variable and its realizations. We consider a class of distributions $\mathcal{P}$ over

$\mathbb{R}^d$ that consists of distributions whose Lebesgue density $p(\cdot)$ is a twice continuously differentiable function. We use $\mathcal{N}(\mu, V)$ to denote a Gaussian distribution of mean $\mu$ and covariance $V$. We use $\|\cdot\|$ to denote the Euclidean norm. For a joint density function $p(\cdot)$ of $(Y, Z)$, let $p_{Z|Y}$ denote $(y, z) \mapsto p(z \mid y)$, a function of both $y$ and $z$.

Suppose $p$ denotes the true-data generating distribution that is usually unknown. Given observed data $y_1, \ldots, y_n$, a typical machine learning problem is to search for a density function $q$ (usually parameterized) over some space that is the most representative of the data. For that purpose, a measure of difference between probability distributions are needed. The existing literature largely replies on the KL divergence, which is defined by $D_{\mathrm{KL}}(p, q) = \mathbb{E}\{\log p(Y)/\log q(Y)\}$ (to log base $e$), where $p(\cdot), q(\cdot)$ are two probability density functions and $\mathbb{E}$ is with respect to $p$. Note that $D_{\mathrm{KL}}(p, q) = -\mathbb{E}\{\log q(Y)\} + \mathbb{E}\{\log p(Y)\}$ and it is only minimized at $q = p$ (almost everywhere). This implies that $-\mathbb{E}\{\log q(Y)\}$ is minimized at $q = p$. A direct consequence of the above observation is the use of maximum likelihood estimation that minimizes $-\sum_{i=1}^{n} \log q(y_i)$. By the law of large numbers, the estimator $\hat{q}$ can be proved to be close to $p$ for large $n$. A possible alternative to KL divergence is the following. The Fisher divergence from a probability density function $q(\cdot)$ to another $p(\cdot)$ is

$$D_{\mathrm{F}}(p, q) = \frac{1}{2} \int_{\mathbb{R}^d} \|\nabla_y \log q(y) - \nabla_y \log p(y)\|^2 p(y) dy,$$

where $\nabla$ is the gradient. It is also referred to as generalized Fisher information distance in physics, and has found application in statistical inference (see, e.g., [12]), with the difference that the $\nabla$ operator is with respect to the parameter instead of data. By similar argument as in the KL divergence, the following result was proved in [9].

**Proposition 1.** *Suppose that the following regularity conditions hold: 1) $p(\cdot)$ and $\nabla_y \log q(\cdot)$ on $(-\infty, \infty)$ are continuously differentiable, 2) $\mathbb{E}\|\log \nabla_y p(y)\|^2$ and $\mathbb{E}\|\log \nabla_y q(y)\|^2$ are finite, 3) $\lim_{|y_j| \to \infty} p(y) \cdot \partial_j \log q(y) = 0$, $(j = 1, \ldots, d)$. Then we have*

$$D_{\mathrm{F}}(p, q) = \mathbb{E}\{s_H(y, q)\} + \frac{1}{2}\mathbb{E}\|\nabla_y \log q(y)\|^2. \tag{2}$$

*where $s_H(y, q)$, referred to as the Hyvarinen loss function, is defined in (1).*

Suppose that a set of observations $y_1, \ldots, y_n$ are drawn from some unknown $p$, a sample analog of $\mathbb{E}\{s_H(y, q)\}$ can be used to search for the $q$ from a space of density functions to approximate $p$ (just like the maximum likelihood estimation). Hyvarinen loss function is particularly powerful when the probability density function is only known up to a multiplicative normalization constant. This is because (1) is invariant under a constant multiplication of $p(y)$. There has been an extension to discrete random variables (see, e.g., [13, 14]). We refer to [9, 11] for more details of Hyvarinen loss in the context of i.i.d. and time series settings. Also, an interesting relationship between Fisher divergence and Stein's operator was pointed out in [15], where the Hyvarinen loss is cast as a specific Stein's operator.

## 2.2 Extension of Hyvarinen loss to partially bounded random variables

The definition in (1) only applies to unbounded random variables. Following an extension to nonnegative variables [16], we further extend the Hyvarinen loss to general continuous variables including unbounded, partially-bounded, and fully bounded cases. Suppose that $y_j \in [a_j, b_j]$, where $a_j, b_j$ $(j = 1, \ldots, d)$ may be at infinity (unbounded case). Suppose there exist nonnegative integers $\alpha_j, \beta_j$ such that

$$p(y) \cdot \partial_{y_j} \log q(y) \cdot (y_j - a_j)^{2\alpha_j}(y_j - b_j)^{2\beta_j} \to 0 \tag{3}$$

as $y \to a_j^+$ or $y \to b_j^-$ for all densities $q$ within the specified model class and true data-generating density $p$. In the above condition, when $a_j = -\infty$ (resp. $b_j = \infty$), it is understood that $\alpha_j = 0$, $(y_j - a_j)^{2\alpha_j} = 1$ (resp. $\beta_j = 0$, $(y_j - b_j)^{2\beta_j} = 1$). Note that the assumption made in [9] corresponds to the unbounded case with $\alpha_j = \beta_j = 0$. For any two vectors $u, v \in \mathbb{R}^d$, and a vector of nonnegative integers $w \in \mathbb{N}^d$, let $u \circ v$ and $u^\alpha$ denote vectors in $\mathbb{R}^d$ whose $j$th entry is $u_j v_j$, $u_j^{\alpha_j}$, respectively. As a special case, $v^0 = 1$ for any scalar $v$. Clearly, the operation $\circ$ is associative. Let $a, b, \alpha, \beta \in \mathbb{R}^d$, and consider

$$D_{\nabla}(p, q) = \frac{1}{2} \int_{\mathcal{D}_y} \|\{\nabla \log q(y) - \nabla \log p(y)\} \circ \{(y - a)^\alpha\} \circ \{(y - b)^\beta\}\|^2 p(y) dy. \tag{4}$$

Table 1: Examples of gradient entropy versus Shannon entropy for common distributions

| DISTRIBUTION | PARAMETER | DENSITY | SUPPORT | G-ENTROPY | S-ENTROPY |
|---|---|---|---|---|---|
| GAUSSIAN | MEAN $\mu$, VARIANCE $\sigma^2$ | $\frac{1}{\sqrt{2\pi}\sigma}e^{-(y-\mu)^2/2\sigma^2}$ | $(-\infty, \infty)$ | $-\frac{1}{2\sigma^2}$ | $\frac{1}{2}\log(2\pi e\sigma^2)$ |
| GAMMA | SHAPE $\alpha$ AND RATE $\beta$ | $\frac{\beta^\alpha}{\Gamma(\alpha)}y^{\alpha-1}e^{-\beta y}$ | $[0, \infty)$ | $-\frac{1}{2}(\alpha+1)$ | $\alpha + \log\frac{\Gamma(\alpha)}{\beta} + (1-\alpha)\psi(\alpha)$ |
| EXPONENTIAL | RATE $\lambda$ | $\lambda e^{-\lambda y}$ | $[0, \infty)$ | $-1$ | $1 - \log(\lambda)$ |
| UNIFORM | RANGE $a, b$ | $\frac{1}{b-a}$ | $[a, b]$ | $0$ | $\log(b-a)$ |
| PARETO | SCALE $a$, SHAPE $\gamma$ | $\frac{\gamma a^\gamma}{y^{\gamma+1}}$ | $[a, \infty)$ | $-\frac{1+\gamma}{2+\gamma}$ | $\log\frac{a}{\gamma} + 1 + \frac{1}{\gamma}$ |

**Theorem 1.** $D_\nabla(p, q)$ *defined as above equals zero if and only if $p$ equals $q$ almost everywhere. Moreover, assume condition (3) holds, $p(\cdot)$ and $\nabla_y \log q(\cdot)$ are continuously differentiable, and*

$$\max_{1\leq j\leq d} \mathbb{E}|y^{\alpha_j+\beta_j}\partial_j \log h(y)|^2 < \infty, \quad \text{for } h = p, q.$$

*Then $D_\nabla(p, q)$ can be written as $s_\nabla(y, q) + c_p$, where $c_p$ is a constant that only depends on $p$, and $s_\nabla(y, q)$ is defined as*

$$\frac{1}{2}\|\{\nabla \log q(y)\} \circ \{(y-a)^\alpha\} \circ \{(y-b)^\beta\}\|^2 + \sum_{j=1}^d \partial_{y_j}\left\{(\partial_{y_j} \log q(y))(y_j - a_j)^{2\alpha_j}(y_j - b_j)^{2\beta_j}\right\}.$$

The regularity conditions made in Theorem 1 are mild and hold for many commonly used distributions such as sub-Gaussian and sub-exponential families. By Theorem 1, the extended $s_\nabla(y, q)$ inherits the properties of (1) and is applicable to a wide range of continuous random variables. To summarize its desirable properties, $s_\nabla(y, q)$ is

(P1) only a function of $y, \nabla q(y), \nabla^2 q(y)$, which usually has an analytic form to evaluate (for parametric $q$);

(P2) invariant under scaling of $q$, which can be quite favorable when the parameterized density $q$ is known up to a normalizing constant;

(P3) statistically proper [1] in the sense that its expectation is only minimized at $q = p$ (almost everywhere) where $q$ is the true data generating density;

(P4) applicable to a wide range of continuous random variables whose entries may be bounded or unbounded or a mixture of them.

### 2.3 Information quantities and properties
The classical information quantities largely rely on KL divergence. For instance, the Shannon entropy is the expectation of the log loss function, and the mutual information is the KL divergence between the product of marginal densities and joint density. Likewise, starting from the Hyvarinen loss, we define the following entropy, conditional entropy and mutual information that are generally referred to as gradient information.

**Definition 1** (Gradient information)**.** *For a continuous random variable $Y = [Y_1, Y_2]$ we define the following information quantities:*

- *Entropy:* $H_\nabla(Y) = \mathbb{E}\{s_\nabla(Y, p_Y)\}$;

- *Conditional entropy:* $H_\nabla(Z \mid Y) = E_p\{s_\nabla([Y, Z], p_{Z|Y})\}$;

- *Mutual information:* $I_\nabla(Y, Z) = D_\nabla(p_{YZ}, p_Y p_Z)$.

*where $p_Y, p_Z$ denotes the marginal densities and $p_{YZ}$ denotes the joint density of variables $Y, Z$.*

For partially bounded random variables in Subsection 2.2, we let $\alpha, \beta$ be the smallest nonnegative integers such that (3) holds. The above definition can be extended to discrete random variables but we leave this extension to a future work. The gradient entropy (denoted by 'G-entropy') along with the Shannon entropy ('S-entropy') for some common distribution families are tabulated in Table 1.

Let $q = p$ in Theorem 1, it is not difficult to observe the following identity.

$$H_\nabla(Y) = -\frac{1}{2}J(Y) \tag{5}$$

where $J(Y) = \|\{\nabla \log p(y)\} \circ \{(y-a)^\alpha\} \circ \{(y-b)^\beta\}\|^2$. For unbounded $Y$ and zero vectors $\alpha, \beta$, $J(Y)$ reduces to $\int_{\mathbb{R}} p(y)\|\nabla_y \log p(y)\|^2 dy$ which is sometimes called the Fisher information of $Y$ that has many implications in physics (see, e.g. [17]). A related but different definition of Fisher information is the variance of the partial derivative with respect to the parameter of a log-likelihood function. As a consequence of (5), we have $H_\nabla(Y) \leq 0$. Its interpretation is elaborated in Subsection 2.4. Also, $I_\nabla(Y, Z)$ equals to zero if and only if $Y$ and $Z$ are independent.

Next we show that the above information quantities enjoy desirable quantities such as chain rule and conditioning reduces entropy that are reminiscent of the properties of Shannon Information. However, they can be more suitable for machine learning due to computational and interpretation advantages we shall point out.

**Theorem 2.** *Suppose the assumptions in Theorem 1 hold. We have the chain rules*

$$I_\nabla(Y; Z) = H_\nabla(Y) + H_\nabla(Z) - H_\nabla(Y, Z) \qquad (6)$$

$$H_\nabla(Y, Z) = H_\nabla(Y) + H_\nabla(Z \mid Y) \qquad (7)$$

As a by product of Theorem 2, $I_\nabla(Y, Z) = H_\nabla(Z) - H_\nabla(Z \mid Y) \geq 0$ (i.e. conditioning reduces entropy), with equality if and only if $Y, Z$ are independent.

We may also define the following "generalized association":

$$I_c(Y; Z) = -\frac{I_\nabla(Y; Z)}{H_\nabla(Y, Z)} \qquad (8)$$

between two random variables $Y, Z$. It can be proved that $I_c \in [0, 1)$, and $I_c = 0$ if and only if $Y, Z$ are independent. In the bivariate Gaussian case, $I_c = \rho^2$ where $\rho$ is the usual correlation.

It is worth mentioning that not all properties of gradient information are counterparts of those in classical Shannon information. Examples are given in the following proposition that are used in proving our results in the supplementary material.

**Proposition 2.** *For any two unbounded random variables $Y, Z$ whose joint distribution exists and satisfies conditions of Proposition 1, we have*

$$H_\nabla(Z \mid Y) \leq \mathbb{E}\{H_\nabla(Z \mid Y = y)\} \qquad (9)$$

*where the expectation on the right hand side is with respect to $Y$.*

*Suppose further that $Y$ and $Z$ are independent, then*

$$H_\nabla(Y + Z \mid Z) = 2H_\nabla(Y). \qquad (10)$$

The Shannon differential entropy of a random variable measures its descriptive complexity. The Fano's inequality also gives lower bounds of model selection or message decoding error, where a larger bound implies more complexity in description. In contrast, the entropy and conditional entropy in Definition 1 measure the uncertainty in prediction. The following Proposition 3 serves as a continuous analog of Fano's inequality that bounds the mean-squared prediction error, and is also intimately related to the Cramér-Rao bound. It can be extended to multidimensional case but we do not pursue the details here.

**Proposition 3.** *Suppose that $Y$ is an unbounded scalar random variable to predict, and $X$ is a variable (providing side information about $Y$) such that the joint distribution of $(X, Y)$ exists. Suppose that $\hat{Y}(X)$ is any estimate of $Y$ which is only a function of $X$. Then the expected $\mathbb{L}^2$ prediction error satisfies*

$$\mathbb{E}(Y - \hat{Y}(X))^2 \geq \frac{1}{-2H_\nabla(Y \mid X)} \qquad (11)$$

*with equality if and only if $Y$ is Gaussian and independent with $X$, and $\hat{Y}(X) = \mathbb{E}Y$.*

### 2.4  Stability interpretation

In modern machine learning systems with uncertainty in data generating processes, stability is a key issue of concern. We introduce a relationship between the gradient information and KL divergence based information that is widely used in machine learning. For brevity, we narrow our scope to unbounded continuous random variables and introduce the following definition and results. Its proof follows from the de Bruijn's identity, Theorem 1 of [18], and Theorem 2 in Section 2.

---
**Algorithm 1** Generic tree approximation based on gradient information
---
**input** Observations of $Y_1, \ldots, Y_p$
**output** A first-order dependence trees $\tau$, and (optionally) a joint density $p(y_1, \ldots, y_p)$ built on $\tau$
1: Estimate $I_\nabla(Y_i, Y_j)$ for each $i \neq j, i, j = 1, \ldots, p$.
2: Build an undirected weighted graph with $p$ vertices representing $Y_1, \ldots, Y_p$, where the weight between vertices $i, j$ is $I_\nabla(Y_i, Y_j)$.
3: Apply Kruskal's algorithm [21] (or alternative algorithms) to obtain a maximum spanning tree $\tau$.
4: Derive or approximate the conditional distribution $p(y_i \mid y_{j(i)})$ that corresponds to each edge $(i, j(i))$, and thus obtain $p(y_1, \ldots, y_p)$.
---

**Definition 2** (Perturbed random variable). *Let $Y$ be any random variable with a finite variance with density $p(\cdot)$. Let $e$ be an standard Gaussian random variable independent of $Y$. Let $Y_v = Y + \sqrt{v}e$ with density $p_v(\cdot)$.*

**Proposition 4.** *We have*

$$H_\nabla(Y) = -\frac{d}{dv}H(Y_v)\mid_{v=0}, \quad D_\nabla(p, q) = -\frac{d}{dv}D_{\mathrm{KL}}(p_v, q_v)\mid_{v=0}$$

$$H_\nabla(Z \mid Y) = -\frac{d}{dv}H(Z_v \mid Y_v)\mid_{v=0} \quad I_\nabla(Y; Z) = -\frac{d}{dv}I(Y_v; Z_v)\mid_{v=0}$$

The identities in Proposition 4 indicates that gradient information describes the non-equilibrium dynamics of the its counterpart under KL divergence. It further indicates that minimizing gradient information is intimately related to enhancing stability. We refer to [18, 19] for more discussions on this. Moreover, the non-negativeness of $I_\nabla(Y; Z)$ means that perturbing $Y$ and $Z$ with independent noises will decrease their mutual information. Similar interpretations apply to other three identities.

## 3 Applications of Gradient Information

### 3.1 Tree approximation of joint distributions

Many machine learning tasks involve high dimensional data where the number of observations is large compared to the number of variables. One approach that we often undertake is divide and conquer (e.g. grouping different dimensions in smaller subsets, removing irrelevant connections). We then bootstrap the models we learn for the subsets or sparse graphs to the entire data dimensions. An effective approach is to assume a tree dependence structure among the variables to simplify the learning, compress data, or to find good initial states for more complex graphical models. In light of the properties (P1)-(P4) of $s_P(y, p)$ (in Subsection 2.2), we expect to develop faster and more stable algorithms for approximating high-dimensional data distributions based on gradient information. In particular, given an $n$th-order probability distribution $p(X_1, \ldots, X_n)$ with $X_i$ being continuous random variables, we wish to find the following optimal first-order dependence tree $p_\tau$.

**Definition 3.** *A distribution $p_{\tau_0}(X_1, \ldots, X_n)$ ($\tau_0 \in T_n$) follows the optimal first-order dependence tree, if $D_\nabla(p, p_{\tau_0}) \leq D_\nabla(p, p_\tau)$ for all $\tau \in T_n$, where $T_n$ is the set of all possible first-order dependence trees (see Fig. 1a for an illustration).*

An analogous definition is given in the pioneering work of Chow [20], but we use $D_\nabla$ instead of $D_{\mathrm{KL}}$ here to measure the discrepancy of approximation. Exhaustive search is a not computationally feasible for any moderately large $n$, since there are $n^{n-2}$ dependence trees in $T_n$ (by Cayley's formula). Motivated by the Chow-Liu algorithm [20] (based on the KL divergence), we provide the following theorem that enables tree approximation of joint distributions using a fast greedy algorithm with $O(n)$-complexity. The problem is formulated as the search for a maximum spanning tree. The procedure is outlined in Algo. 1.

**Theorem 3.** *Under the assumptions made in Theorem 2, the best tree approximation of a joint distribution $p(y_1, \ldots, y_n)$ under Fisher divergence $D_\nabla(\cdot, \cdot)$ is the maximum spanning tree with weight $I_\nabla(Y_i, Y_{j(i)})$, where $(i, j(i))$ is an edge that denotes $p(Y_i \mid Y_{j(i)})$.*

We apply our algorithm to a protein signaling flow cytometry dataset. The dataset encodes the presence of $p = 11$ proteins in $n = 7466$ cells. It was first analyzed using Bayesian networks in [22] who fit a directed acyclic graph to the data, later studied in [23] using different methods. The tree can be used as a graphical visualization tool to highlight the most highly correlated genes in

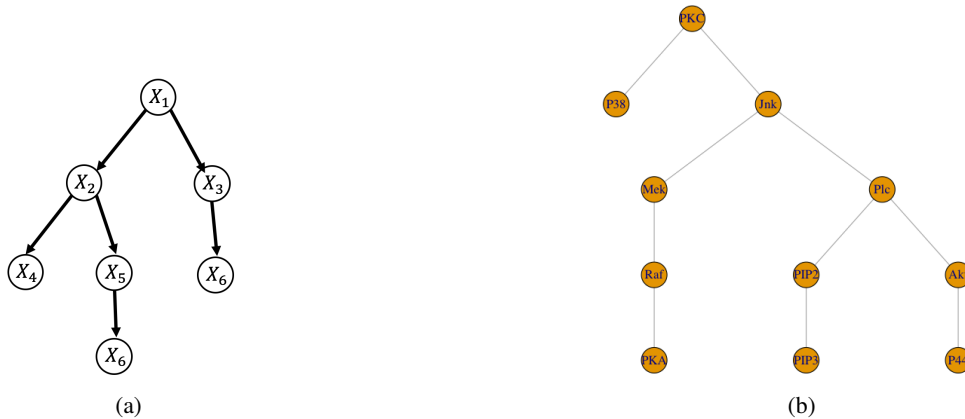

(a)

(b)

Figure 1: (a) A joint distribution with tree structure $p(x_1, \ldots, x_6) = p(x_1)\, p(x_2 \mid x_1)\, p(x_4 \mid x_2)$ $p(x_5 \mid x_2)\, p(x_6 \mid x_5)\, p(x_3 \mid x_1)\, p(x_6 \mid x_3)$, and (b) the tree discovered from the protein data.

the correlation network. We suppose that any pair of random variables $Y_1, Y_2$ follow the following exponential family distribution $p(y_1, y_2) \propto \exp\{\theta_1 y_1^2 y_2^2 + \theta_2 y_1^2 + \theta_3 y_2^2 + \theta_4 y_1 y_2 + \theta_5 y_1 + \theta_6 y_2\}$. Note that the constant is a function of $\theta$ and it does not have a closed form. The above distribution is a special case of a class of exponential family distributions with normal conditionals. This family is intriguing from the perspective of graphical modeling as, in contrast to the Gaussian case, conditional dependence may also express itself in the variances [23]. To estimate the density, we minimize the sample average of (1), and obtain a closed form solution $\hat{\theta}$ (to be elaborated in the supplement). Based on the estimates, we can obtain a consistent estimator of the entropy $H_\nabla(Y_1, Y_2)$ by a sample analog of (5), using Monte Carlo samples generated from the estimated density. To calculate $H_\nabla(Y_1)$ and $H_\nabla(Y_2)$, we calculate the marginal distributions in closed form, and obtain a consistent estimation of entropy and mutual information by sample analogs. The details are elaborated in the "derivations for exponential family example" section in the supplement. Figure 1b shows the network structure after applying our method to the data using the proposed approach. Our result is consistent with the estimated graph structure in [22]. We record the computational time by running different number of variables ($p$ from 2 to 11) in Fig 2a, which shows the gradient information based algorithm is more than 100 times faster than Shannon information based algorithm.

It is worth noting that the method of Algo. 1 can also be used to perform supervised classification. Given features $X_1, \ldots, X_p$ and their corresponding label $Y$, we calculate the tree distribution that approximates the joint distribution of $p(x_1, \ldots, x_p)$ for each class of $Y$. We then perform a likelihood

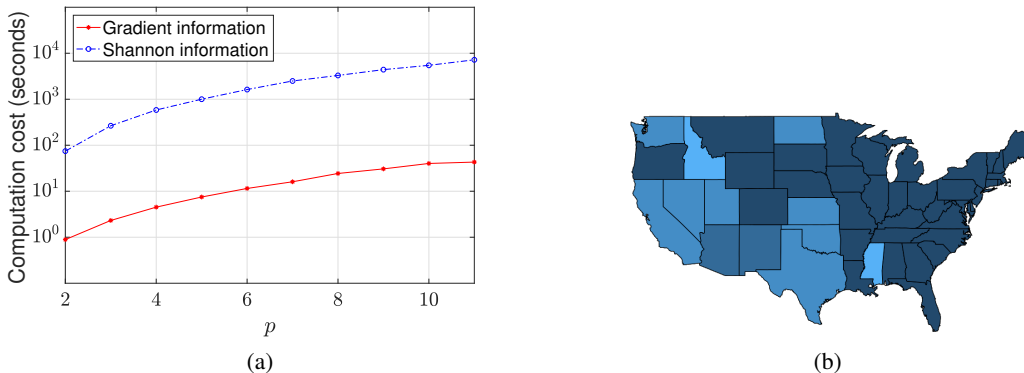

(a)

(b)

Figure 2: (a) Comparison of computational costs of Chow-Liu tree approximation using classical Shannon information [20] and gradient information (proposed here), depicted in logarithmic scale; (b) The communities of states detected from quarterly growth rates of payroll employment for the U.S. states (the number of communities is set to be four).

Table 2: Classification accuracy of three methods for data with different levels of correlation

|  | $\rho = 0.3$ | $\rho = 0.5$ | $\rho = 0.7$ | $\rho = 0.9$ |
|---|---|---|---|---|
| CHOW-LIU TREE | 0.610 | 0.829 | 0.965 | 0.994 |
| RANDOM FOREST | 0.607 | 0.759 | 0.886 | 0.953 |
| ELASTIC NET | 0.535 | 0.537 | 0.541 | 0.526 |

---

**Algorithm 2** Community discovery based on mutual information

---

**input** Observations of $Y_1, \ldots, Y_p$, number of communities $k$ ($1 \leq k \leq p$)
**output** A partition of $\{1, \ldots, p\}$ into $k$ subsets
1: Apply Algo. 1 to obtain the spanning tree $\tau$ (with weights being the mutual information).
2: Remove the $k$ edges that have the smallest weights from $\tau$.
3: The output partition is represented by the connected components of the current $\tau$.

---

ratio test to decide which class a given feature vector $x_1, \ldots, x_p$ is associated with. In calculating the joint density, we let the spanning tree be rooted at a node with the largest geodesic distance and the rest of the nodes are ranked according to the height directed preorder (HDP) traversal of the tree [24]. In a synthetic data experiment, we generate two classes of data from an independent Gaussian vector $[X_1, \ldots, X_p]$. The covariance $\text{Cov}(X_i, X_j)$ of the first class of data is $\rho^{|i-j|}$, and the covariance of the second class is $(-\rho)^{|i-j|}$ (for $i, j = 1, \ldots, p$). We generated 100 data in each of the 1000 replications and record the cross validation accuracy (with $30\%$ test data) in Table 2. We also compared our method (denoted by "Chow-Liu tree") with two popular classification methods, random forest [25] and elastic net [26]. The results indicate the superior performance of our method. Elastic net does not work well mainly because the two classes of data here are not linearly separable.

## 3.2 Community discovery

Many real-world networks of data exhibit a community structure: the vertices of the network are partitioned into groups such that the statistical dependence is high among vertices in the same group and low otherwise. Most of the community detection methods (e.g. stochastic block models [27]) focus on the concentration of linkages in random graphs. Here we provide an alternative perspective using gradient mutual information. More precisely, we do not perform communities discovery from edge connections, but from dependence among variables/vertices (assuming multiple observations at each vertex). Such dependence is quantified by mutual information. And the obtained communities can be understood as disjoint subsets of variables that exhibit large within-community dependence and small inter-community dependence. We propose a fast community discovery approach based on Algo. 1. The main idea, summarized in Algo. 2, is to first obtain a spanning tree that best represents the joint distribution, and then construct communities by removing weak-dependence connections.

In a data study, we considered a dataset constructed in [28]. The data was also studied in [29] using an algorithm that recovers the communities using the eigenvectors of the sample covariance matrix. The data consists of quarterly growth rates of payroll employment for the U.S. states (excluding Alaska and Hawaii) from the second quarter of 1956 to the fourth of 2007, which results in a panel of $n = 48$ time series over $T = 207$ periods. The data are seasonally adjusted and annualized. We show the results of applying our clustering algorithm to the sample in Figure 2b. The communities roughly match the clusters of Fig. 3 in [29] using a partial correlation network model.

## 4 Conclusion

Most existing methods for determining randomness, conditional randomness, causality, discrepancy, information gains, etc. in data representation and modeling largely depend the on KL divergence and logarithmic loss. To facilitate interpretability and reduce computation, at least in some occasions, we introduced gradient information as an alternative to classical information measures widely used in machine learning. A future work is to apply the gradient information to community detection with more complicated graph structures instead of tree structures. Another future work is to apply gradient information to evaluate information gains and information bottleneck with a reduced computational complexity. An anonymous reviewer also pointed out the possibility of defining cross entropy as another analogy to the classical KL divergence based cross entropy.

**Acknowledgments**

This work was supported by DARPA Grant No. HR00111890040.

**Supplementary material**

The supplementary material includes proofs of theoretical results, and two additional applications of gradient information. One application is about channel stability. The second application is in derivation of some general inequalities using gradient information (which are otherwise highly nontrivial to establish).

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
