[Supplementary Material]

# Supplementary Material for
# "Gradient Information for Representation and Modeling"

## Supplementary Material

In the supplement, we include proofs of theoretical results, and provide three additional applications of gradient information. One application is on Granger-type causality measures; The second application is about channel stability in information theory; The last application is in derivation of some general inequalities using gradient information (which are otherwise highly nontrivial to establish).

## Proof of Theorem 1

It can be seen that $D_\nabla(p,q) = 0$ if and only if $\nabla p(y) = \nabla q(y)$ almost everywhere. Since $p$ and $q$ are densities and integrate to one, $\nabla p(y) = \nabla q(y)$ is equivalent to $p(y) = q(y)$ almost everywhere.

Using direct calculation and integration by parts, we have

$$D_\nabla(p,q) = \frac{1}{2}\int_{[a,b]} p(y)|(\nabla \log q(y)) \circ (y-a)^{\alpha_j} \circ (y-b)^{\beta_j}|^2 dx$$
$$-\sum_{j=1}^{d}\int_{[a_j,b_j]} p(y)\,(\partial_{y_j}\log p(y))(\partial_{y_j}\log q(y))(y_j-a_j)^{2\alpha_j}(y_j-b_j)^{2\beta_j}\,dy_j + C$$

and the $j$-th term in the above summation is

$$-\int_{[a_j,b_j]} p(y)\,(\partial_{y_j}\log p(y))(\partial_{y_j}\log q(y))(y_j-a_j)^{2\alpha_j}(y_j-b_j)^{2\beta_j}\,dy_j$$
$$= -\int_{[a_j,b_j]}(\partial_{y_j}p(y))\,(\partial_{y_j}\log q(y))(y_j-a_j)^{2\alpha_j}(y_j-b_j)^{2\beta_j}\,dy_j$$
$$= -p(y)(\partial_{y_j}\log q(y))(y_j-a_j)^{2\alpha_j}(y_j-b_j)^{2\beta_j}\,|_{a_j}^{b_j} + \int_{[a_j,b_j]} p(y)\,\partial_{y_j}\left\{(\partial_{y_j}\log q(y))(y_j-a_j)^{2\alpha_j}(y_j-b_j)^{2\beta_j}\right\}dy_j$$
$$= \int_{[a_j,b_j]} p(y)\,\partial_{y_j}\left\{(\partial_{y_j}\log q(y))(y_j-a_j)^{2\alpha_j}(y_j-b_d)^{2\beta_j}\right\}dy_j$$

where the last identity holds under (3).

## Proof of Theorem 2

We use $\nabla_{y,z}$ and $\nabla_y$ to highlight that the derivative is taken with regard to $[y,z]$ and $y$, respectively. We only prove for unbounded $Y, Z$. The proof of the extended case is similar, as discussed in Subsection 2.2.

We first prove Identity (6). Applying Proposition 1 and Identity (5), we obtain the following identities (where expectations are with respect to $p_{YZ}$):

$$D_\nabla(p_{YZ}, p_Y p_Z) = \mathbb{E}\{s_\nabla([Y,Z], p_Y p_Z)\} + \frac{1}{2}\mathbb{E}\|\nabla_{y,z}\log p_{Y,Z}(Y,Z)\|^2$$
$$= \frac{1}{2}\mathbb{E}\left(\|\nabla_y\log\{p_Y(Y)p_Z(Z)\}\|^2 + \|\nabla_z\log\{p_Y(Y)p_Z(Z)\}\|^2\right)$$
$$+ \Delta_y\log\{p_Y(Y)p_Z(Z)\} + \Delta_z\log\{p_Y(Y)p_Z(Z)\} - \mathbb{E}\{s_\nabla([Y,Z], p_{YZ})\}$$
$$= \mathbb{E}\{s_\nabla(Y, p_Y)\} + \mathbb{E}\{s_\nabla(Z, p_Z)\} - \mathbb{E}\{s_\nabla([Y,Z], p_{YZ})\}.$$

367 We then prove Identity ([7](#)). Direct calculations give

$$
\begin{aligned}
\mathbb{E}\{s_\nabla([Y,Z], p_{Y,Z})\} &= \frac{1}{2}\mathbb{E}\|\nabla_{y,z}\{\log p_Y(Y) + \log p_{Z|Y}(Z\mid Y)\}\|^2 + \mathbb{E}\big(\Delta_{y,z}\{\log p_Y(Y) + \log p_{Z|Y}(Z\mid Y)\}\big)\\
&= \frac{1}{2}\mathbb{E}\big(\|\nabla_y \log p_Y(Y)\|^2 \\
&\quad + \|\nabla_{y,z}\log p_{Z|Y}(Z\mid Y)\|^2\big) + \mathbb{E}\{\Delta_y \log p_Y(Y)\} + \mathbb{E}\{\Delta_{y,z}\log p_{Z|Y}(Z\mid Y)\} + c \\
&= H_\nabla(Y) + H_\nabla(Z\mid Y) + c
\end{aligned}
$$

368 where $c$ denotes

$$
c = \mathbb{E}\{\nabla_y \log p_Y(Y)^\mathrm{T} \cdot \nabla_y \log p_{Z|Y}(Z\mid Y)\}.
$$

369 It remains to show that $c = 0$. We use $d_y$, $\mathcal{D}_y$, $\mathcal{D}_{y_{(-j)}}$ to denote the dimension of $Y$, domain of
370 $Y$, domain of the subvector of $Y$ excluding dimension $j$, respectively. $\mathcal{D}_z$ and $\mathcal{D}_{y,z}$ are similarly
371 defined. We have

$$
\begin{aligned}
c &= \mathbb{E}\{\nabla_y \log p_Y(Y)^\mathrm{T} \cdot \nabla_y \log p_{Z|Y}(Z\mid Y)\} \\
&= \sum_{j=1}^{d_y} \int_{\mathcal{D}_{y,z}} p_{Y,Z}(y,z)\frac{\partial_{y_j} p_Y(y)}{p_Y(y)} \cdot \frac{\partial_{y_j} p_{Z|Y}(z\mid y)}{p(z\mid y)} dy dz \\
&= \sum_{j=1}^{d_y} \int_{\mathcal{D}_{y,z}} \partial_{y_j} p_Y(y) \cdot \partial_{y_j} p(z\mid y) dy dz \\
&= \sum_{j=1}^{d_y} \int_{\mathcal{D}_{y_{(-j)},z}} \left( p_Y(y)\cdot \partial_{y_j} p(z\mid y)\mid_{-\infty}^{\infty} - \int_{\mathcal{D}_j} p_Y(y)\partial_{y_j}^2 p(z\mid y) dy_j \right)\prod_{k\neq j} dy_k dz \\
&= -\int_{\mathcal{D}_y} p_Y(y)\left(\int_{\mathcal{D}_z}\Delta_y p(z\mid y)dz\right)dy = -\int_{\mathcal{D}_y} p_Y(y)\left(\Delta_y \int_{\mathcal{D}_z} p(z\mid y)dz\right)dy = 0.
\end{aligned}
$$

## Proof of Proposition [2](#)

373 We first prove ([9](#)). By a derivation similar to the proof of Theorem [2](#), we obtain

$$
H_\nabla(Z\mid Y) = -\frac{1}{2}\mathbb{E}\|\nabla_{[z,y]}\log p(z\mid y)\|^2 \tag{13}
$$

374 for any two random variables $Z, Y$ whose joint distribution exists. Therefore,

$$
H_\nabla(Z\mid Y) = -\frac{1}{2}\mathbb{E}\|\nabla_{Y,Z}\log p_{Z|Y}(Z\mid Y)\|^2 \leq -\frac{1}{2}\mathbb{E}\|\nabla_Z \log p_{Z|Y}(Z\mid Y)\|^2 = \mathbb{E}\{H_\nabla(Z\mid Y=y)\}.
$$

375 We then prove ([10](#)). Suppose $W = Y + Z$. It follows from ([13](#)) that

$$
\begin{aligned}
H_\nabla(W\mid Z) &= -2\times\frac{1}{2}\mathbb{E}\|\nabla_w \log p(w-z)\|^2 \\
&= -\mathbb{E}\|\nabla_y \log p(y)\|^2 = 2H_\nabla(Y).
\end{aligned}
$$

## Proof of Proposition [3](#)

377 **Lemma 1.** *Given a fixed covariance matrix $V$ of a random variable $Y$ supported on $\mathbb{R}^d$, the*
378 *distribution that maximizes $H_\nabla(Y)$ is Gaussian (with an arbitrary mean), and the maximum is*
379 $-Tr(V^{-1})/2$.

380 We now prove that the maximum entropy distribution on $\mathbb{R}^d$ is Gaussian given second moment
381 constraints. The results are readily observable from the known results that the distribution with a
382 fixed variance that minimizes the Fisher information is the Gaussian distribution, typically proved
383 using calculus of variations and differential equations (see, e.g. [[28](#)]). Here we provide a much
384 simpler proof.

*Proof.* Suppose that $Y_1, Y_2$ are two i.i.d. random variables following the maximum entropy distribution. Then $2Y_1$ follows the maximum entropy distribution with variance $2V$, and by definition, $J(2Y_1) \leq J(Y_1 + Y_2)$. Direction calculations show that $J(2Y_1) = J(Y_1)/2$, therefore, it follows from the convolution inequality [29] that

$$\frac{J(Y_1)}{2} = J(2Y_1) \leq J(Y_1 + Y_2) \leq \frac{1}{J(Y_1)^{-1} + J(Y_2)^{-1}} = \frac{J(Y_1)}{2}$$

with equality only if the equality for convolution inequality for Fisher information holds, which implies that $Y_1, Y_2$ must be Gaussian. □

*Proof of Proposition 3:*

By Lemma 1, we have

$$\mathbb{E}(Y - \hat{Y}(X))^2 = \mathbb{E}_X \mathbb{E}_{Y|X}(Y - \hat{Y}(x) \mid X = x)^2 \geq \mathbb{E}_X Var(Y \mid X = x) \geq \mathbb{E}_X \frac{1}{-2H_\nabla(Y \mid X = x)}.$$

Moreover, applying Cauchy's inequality and Identity 9, we obtain

$$\mathbb{E}_X \frac{1}{-2H_\nabla(Y \mid X = x)} \geq \frac{1}{\mathbb{E}_X\{-2H_\nabla(Y \mid X = x)\}} \geq \frac{1}{-2H_\nabla(Y \mid X)}.$$

This concludes the proof.

# Proof of Theorem 3

Let $p^a$ denote a distribution with first-order dependence tree structure. Using Proposition 1 and Identity 5, we have

$$\mathbb{E}\{D_\nabla(p, p^a)\} = \mathbb{E}\{s_\nabla(Y, p^a)\} + \frac{1}{2}\mathbb{E}\|\nabla_y \log p(y)\|^2 = \mathbb{E}\{s_\nabla(Y, p^a)\} - H_\nabla(Y) \tag{14}$$

In order to minimize $E\{D_\nabla(p, p^a)\}$, we only need to minimize $\mathbb{E}\{s_\nabla(Y, p^a)\}$. Let $j(i)$ index the parent node of $i$ on the tree that represents $p^a$. Without loss of generality, let $Y_1$ denote the root of the tree, and $E_a$ denote the set of edges. By Identity (7) in Theorem 2, the Markovity of $p^a$, and the fact that $p^a(\cdot)$ agrees with $p(\cdot)$ on all the first and second order marginal distributions, we can rewrite $\mathbb{E}\{s_\nabla(Y, p^a)\}$ as

$$
\begin{aligned}
\mathbb{E}\{s_\nabla(Y, p^a)\} &= \mathbb{E}\{s_\nabla(Y_1, p_1^a)\} + \sum_{\{i,j(i)\} \in E_a} \mathbb{E}\{s_\nabla(Y_i, p_{i|j(i)}^a)\} \\
&= H_\nabla(Y_1) + \sum_{\{i,j(i)\} \in E_a} \{H_\nabla(Y_i) - I_\nabla(Y_i; Y_{j(i)})\} \\
&= \sum_{i=1}^{n} H_\nabla(Y_i) - \sum_{\{i,j(i)\} \in E_a} I_\nabla(Y_i; Y_{j(i)}).
\end{aligned}
$$

This concludes the proof.

# Derivations for exponential family example

The sample average of (1) is a quadratic function of $\theta = [\theta_1, \ldots, \theta_5]$. The closed form solution $\hat{\theta}$ is derived as

$$\hat{\theta} = -\left\{\sum_{j=1}^{n}(a_{1,j}a_{1,j}^{\mathrm{T}} + a_{2,j}a_{2,j}^{\mathrm{T}})\right\}^{-1} \sum_{j=1}^{n}(a_{3,j} + a_{4,j}) \tag{15}$$

where

$$
\begin{aligned}
a_{1,j} &= [2y_{1,i}y_{2,i}^2, 2y_{1,i}, 0, y_{2,i}, 1, 0]^{\mathrm{T}}, \quad a_{2,j} = [2y_{1,i}^2 y_{2,i}, 0, 2y_{2,i}, y_{1,i}, 0, 1]^{\mathrm{T}}, \\
a_{3,j} &= [2y_{2,i}^2, 2, 0, 0, 0, 0]^{\mathrm{T}}, \quad a_{4,j} = [2y_{1,i}^2, 0, 2, 0, 0, 0]^{\mathrm{T}},
\end{aligned}
$$

The distribution density of $Y_2$ is

$$p(y_2) \propto (-\theta_1 y_2^2 - \theta_2)^{-1/2} \exp\left\{-\frac{(\theta_4 y_2 + \theta_5)^2}{4(\theta_1 y_2^2 + \theta_2)} + \theta_3 y_2^2 + \theta_6 y_2\right\} \qquad (16)$$

So its entropy can be estimated by

$$-\frac{1}{2n}\sum_{j=1}^{n}\|\nabla \log p_{\hat{\theta}}(y_2)\|^2$$

$$= -\frac{1}{2n}\sum_{j=1}^{n}\left\{-\frac{1}{2}\frac{2\hat{\theta}_1 y_{2,j}}{\hat{\theta}_1 y_{2,j}^2 + \hat{\theta}_2} + \frac{1}{4}\frac{2\hat{\theta}_1 y_{2,i}(\hat{\theta}_4 y_{2,i} + \hat{\theta}_5)^2}{(\theta_1 y_{2,i}^2 + \theta_2)^2} - \frac{1}{4}\frac{2\hat{\theta}_4(\hat{\theta}_4 y_{2,i} + \hat{\theta}_5)}{\theta_1 y_{2,i}^2 + \theta_2} + 2\hat{\theta}_3 y_{2,i} + \hat{\theta}_6\right\}^2.$$

The value of $H_\nabla(Y_1)$ can be similarly estimated. We therefore get an consistent (under some moment conditions) estimate $I_\nabla(Y_1, Y_2)$ from Proposition 5.

The conditional distribution density $p(y_1 \mid y_2)$ can be calculated from (12) and (16):

$$p(y_1 \mid y_2) \propto (-\theta_1 y_2^2 - \theta_2)^{1/2} \exp\left\{\theta_1 y_1^2 y_2^2 + \theta_2 y_1^2 + \theta_4 y_1 y_2 + \theta_5 y_1 + \frac{(\theta_4 y_2 + \theta_5)^2}{4(\theta_1 y_2^2 + \theta_2)}\right\} \qquad (17)$$

## Application: Implication on Causality

The identification of causality usually serves as a key step towards simplified modeling and learning. Let $X_1, X_2, \ldots$ and $Y_1, Y_2, \ldots$ be two sequences of data. In general, we say a series $X_t$ causes another series $Y_t$ if knowing the past $\{X_1, \ldots, X_{t-1}\}$ can provide information on the future of $Y_t$ given the past of $\{Y_1, \ldots, Y_{t-1}\}$. This school of thoughts is exemplified by the seminal work of Granger [30] in identifying causal relations between multivariate times series. The Granger causality is typically tested in linear models between $Y_t$ and $X_t$ (with lags) and the two processes are assumed to be stationary [31]. In general, this type of of causality can be unified by Kolmogorov complexity $\mathbb{K}(\cdot)$ which, not only extends Granger causality to nonstationary and nonlinear processes, but also includes various other approximations of Kolmogorov information in the literature, such as Shannon's mutual information, Renyi's information, directed Shannon information, directed Renyi information, combinatorial measures of information (e.g. Lempel-Ziv information).

Intuitively, the quantity measures how much complexity of the series $\{Y_t\}$ is reduced by knowing $\{X_t\}$. The past $\{X_1, \ldots, X_{t-1}\}$ provide information about $Y_t$ if $\mathbb{K}(Y_t|X_{t-1}, \ldots, X_1, Y_{t-1}, \ldots, Y_1) < \mathbb{K}(Y_t|Y_{t-1}, \ldots, Y_1)$ for Kolmogorov complexity measure $\mathbb{K}(\cdot)$. In this case, additional predictive information is provided by the series $\{X_t\}$ if

$$\mathbb{K}(Y_t|Y_{t-1}, \ldots, Y_1) - \mathbb{K}(Y_t|Y_{t-1}, \ldots, Y_1, X_{t-1}, \ldots, X_1)$$

is greater than zero. We define the left hand side of the above term to be the Kolmogorov causal information provided by series $\{X_t\}$ for predicting $\{Y_t\}$.

However, Kolmogorov information is in general not computable and its surrogates may be used instead. For example, consider replacing the complexity measure $\mathbb{K}$ by Shannon entropy $H(Y) = -\int p(y) \log p(y) dy$ for a continuous random variable $Y$. In this case, Kolmogorov causal information reduces to the directed Shannon information [32].

Using gradient entropy as the surrogate for the Kolmogorov information, we can define gradient-directed information (as a surrogate for Kolmogorov causal information) as:

$$H_\nabla(Y_t|Y_{t-1}, \ldots, Y_1) - H_\nabla(Y_t|Y_{t-1}, \ldots, Y_1, X_{t-1}, \ldots, X_1).$$

The above measure provides an alternative method of measuring the causality from the sensitivity point of view. It also provides significant computational advantages particularly when density normalizing constants are unknown.

## Application: Stability of Channel Capacity

Consider input $X$, output $Y$, and a time invariant channel described by conditional distribution $p_{Y|X}$. The channel capacity in many cases is achieved by maximizing $I(X;Y)$ over the marginal

distribution of $X$, $p_X$ (see for example the channel coding theorem [33]). In practical applications, we may also be interested in the stability of channel capacity, in the sense that the capacity is not very sensitive to perturbations at both ends of the channel. One possible way to define the channel stability is through the definition of $I_\nabla(X;Y)$ which can be interpreted as the sensitivity of $I(X;Y)$: the smaller the better (see Subsection 2.4).

It is thus reasonable to assume that solutions of the following types of optimization problem would lead to channel coding that is both efficient in transmitting signals and robust against noise.

$$\max_{p_X} I(X;Y)/I_\nabla(X;Y). \tag{18}$$

It has been well known that Gaussian input maximizes the Gaussian channel capacity under power constraints. In the following theorem, we show that Gaussian input also minimizes $I_\nabla(X;Y)$ for Gaussian channels under power constraints. The result further indicates that Gaussian input achieves the optimum of (18), i.e. it enjoys *not only the largest information capacity but also the smallest instability*.

**Theorem 4.** *For Gaussian channel $Y = X + N$ where $N \sim \mathcal{N}(0, v_N)$ and $var(X) = v_X$, the $p_X$ that achieves the minimum of $I_\nabla(X;Y)$ is Gaussian.*

$$\min_{p_X} I_\nabla(X;Y). \tag{19}$$

*Moreover, the smallest mutual information is*

$$I_\nabla(X;Y) = H_\nabla(Y) - H_\nabla(Y \mid X) = -\frac{1}{2(v_X + v_N)} + \frac{1}{v_N} = \frac{2v_X + v_N}{2(v_X + v_N)v_X},$$

*which is increasing in $v_X$ with range $[v_N^{-1}, 2v_N^{-1})$.*

*Proof.* By Identity (7) in Theorem 2, we have $I_\nabla(X;Y) = H_\nabla(Y) - H_\nabla(Y \mid X)$. Using Identity 10, we have

$$H_\nabla(Y \mid X) = H_\nabla(X + N \mid X) = 2H_\nabla(N) = -\frac{1}{v_N}$$

which is a constant that does not depend on $X$. Thus, minimizing (19) is equivalent to minimizing $H_\nabla(Y)$ under the constraint that $Var(Y) \le v_N + v_X$. Using Lemma 1 concludes the proof. $\square$

## Application: Some Elementary Inequalities

**Proposition 5.** *Under the same assumptions of Theorem 2, we have*

$$H_\nabla(Y_1, \ldots, Y_n) = \sum_{i=1}^{n} H_\nabla(Y_i \mid Y_1, \ldots, Y_{i-1}) \le \sum_{i=1}^{n} H_\nabla(Y_i). \tag{20}$$

*Proof.* The result can be obtained by recursively applying Identity (7) in Theorem 2 for $n$ random variables $Y_1, \ldots, Y_n$. $\square$

We can generalize Proposition 5 to show how the monotonicity of the average entropy rates of subsets as the size of the subsets increases.

**Proposition 6.** *Suppose that $Y_1, \ldots, Y_n$ have a joint distribution. For every $S \subseteq \{1, \ldots, n\}$, let $Y_S = \{Y_i : i \in S\}$, and $Y_{S^c} = \{Y_i : i \notin S\}$. Let*

$$h_k^{(n)} = \frac{1}{\binom{n}{k}} \sum_{S:card(S)=k} \frac{H_\nabla(Y_S)}{k}, \tag{21}$$

$$t_k^{(n)} = \frac{1}{\binom{n}{k}} \sum_{S:card(S)=k} \frac{1}{-H_\nabla(Y_S)/k}, \tag{22}$$

*Then*

$$h_1^{(n)} \ge \cdots \ge h_n^{(n)}$$
$$t_1^{(n)} \ge \cdots \ge t_n^{(n)}$$

467 *Proof.* Inequalities ([21](#)) can be proved using the same arguments as in the proof of Theorem 16.8.1
468 in [[33](#)] (which only uses Proposition [5](#)). We only prove the Inequality ([22](#)) here. The proof of
469 Theorem 16.8.1 in [[33](#)] implies that

$$\frac{1}{n} \sum_{i=1}^{n} \frac{H_\nabla(Y_{-i})}{n-1} \geq \frac{1}{n} H_\nabla(Y_1, \ldots, Y_n) \tag{23}$$

470 where $Y_{-i}$ denotes $[Y_1, \ldots, Y_{i-1}, Y_{i+1}, Y_n]$. Thus, we have

$$\frac{1}{n} \sum_{i=1}^{n} \frac{1}{\frac{-H_\nabla(Y_{-i})}{n-1}} \geq \frac{1}{\frac{1}{n}\sum_{i=1}^{n} \frac{-H_\nabla(Y_{-i})}{n-1}} \geq \frac{1}{-\frac{1}{n}H_\nabla(Y_1, \ldots, Y_n)} \tag{24}$$

471 where the first inequality of ([24](#)) follows from the fact that $H_\nabla(\cdot)$ is a negative function and the
472 harmonic mean is no larger than the arithmetic mean, and the second inequality follows from ([23](#)).
473 Inequality ([24](#)) is equivalent to $t_{n-1}^{(n)} \geq t_n^{(n)}$. To prove ([22](#)), we first condition on a $k$-element subset,
474 and apply the existing result to obtain $t_{k-1}^{(k)} \geq t_n^{(k)}$. We then take a uniform choice over the $k$-element
475 subset and its $k-1$-element subsets. $\qquad\square$

Consider the specific case where $Y_i$'s are jointly distributed according to Gaussian distribution with
covariance $V$. We can have inequalities for traces, as Gaussian gradient entropy is

$$-\frac{1}{2}\text{Tr}(V^{-1}).$$

476 Throughout the remainder of this chapter, we will assume that $V$ is a positive definite symmetric
477 $n \times n$ matrix.

478 **Proposition 7.** *For any positive definite matrix $V$, we have*

$$Tr(V^{-1}) \geq \sum_{i=1}^{n} Tr(V_i^{-1}). \tag{25}$$

479 *for any set of block matrices $\{V_1, \ldots, V_n\}$ of $V$ whose rows (resp. columns) form a partition of the*
480 *rows (resp. columns) of $V$. Moreover, the equality holds if and only if $V$ are block-diagonal with*
481 *blocks $\{V_1, \ldots, V_n\}$.*

482 Inequality ([25](#)) immediately follows from ([20](#)). We note that the inequality in ([25](#)) can also be
483 proved by using block matrix inversion and Woodbury matrix identity, but it is much more involved
484 compared with the simple proof here using entropy inequality.

485 **Proposition 8.** *If $h_k$ denotes the product of the determinants of all the principal $k$-rowed minors of*
486 *a positive definite $n \times n$ matrix $V$, i.e.,*

$$h_k = \sum_{1 \leq i_1 < \cdots < i_k \leq n} Tr(V(i_1, \ldots, i_k)^{-1})$$

487 *then*

$$h_1 \leq \cdots \leq \frac{h_k}{\binom{n-1}{k-1}} \leq \cdots \leq h_n$$

488 *with equality if and only if $V$ is a diagonal matrix.*

489 *Proof.* Let $X \sim \mathcal{N}(0, V)$. Then the inequality follows directly from Proposition [6](#) and $k\binom{n}{k} = $
490 $n\binom{n-1}{k-1}$. The proof of Inequality ([21](#)) implies that if $X$ is Gaussian, the equality holds if and only if
491 the entries of $X$ are independent, i.e., $V$ is diagonal. $\qquad\square$