[Reviews · NeurIPS 2019]

Reviewer 1



The Hyvarinen scoring function seems to be a special case of the Stein score function (see Eq. (1) in Liu, Lee & Michael Jordan. "A kernelized Stein discrepancy for goodness-of-fit tests." ICML 2016. and set f(x) to be the gradient). As is pointed out in the ICML 2016 paper, it would be better to allow f(x) to be selected from a set of smooth functions (e.g. functions from the reproducing kernel Hilbert Schdmit space) so that minimizing the resulting discrepancy measure can push the density function to match the empirical data distribution. Would the proposed extensions still hold for the Stein score function? Could the author elaborate on the statement that "compared with Shannon differential entropy of a random variable that measures it descriptive complexity, the entropy and conditional entropy defined in Definition 1 measure the uncertainty in prediction complexity"? Wouldn't the Fano's inequality also characterize the prediction complexity? Overall, this work can be improved by adding more discussions on existing works on the Stein score function listed below: [1] Liu, Qiang, Jason Lee, and Michael Jordan. "A kernelized Stein discrepancy for goodness-of-fit tests." ICML 2016. [2] Chwialkowski, Kacper, Heiko Strathmann, and Arthur Gretton. "A kernel test of goodness of fit." ICML 2016. [3] Gorham, Jackson, and Lester Mackey. "Measuring sample quality with kernels." ICML 2017. [4] Liu, Qiang, and Dilin Wang. "Stein variational gradient descent: A general purpose bayesian inference algorithm." NIPS 2016.

Reviewer 2



This paper develops new information quantities. These quantities are derived from the Hyvarinen loss and are shown to be related to Fisher divergence. The behavior of some of these quantities is similar in part to classical information quantities such as entropy, mutual information, KL divergence etc. The former quantities are said to be faster to compute and more robust than the latter. In this paper, these quantities are derived, studied and used to derive several learning algorithms, most notably a version of the Chow-Liu algorithm based on these quantities. The resulting algorithm is used for structure learning, classification and community detection. The contribution is imho rather original. Learning graphical models over continuous variables receive some attention, works often limit themselves to Gaussian variables. I think the paper is well written, though it should be proofread to remove all remaining typos. It is also well organized and the contributions are clear. I think the authors did a great job moving the proofs and details to the appendix. I think the paper is moderately significant. Learning graphical models over continuous variables has always been challenging. The proposed information quantities have the potential to provide a well-founded and computationally interesting solution to this problem, which sounds very exciting to me. The experiment also clearly illustrates the computational advantages of the proposed quantities. However and as far as I understand, the paper only discusses pairs of variables and the authors postpone extension to more variables for later works, so I was a bit disappointed. Computational challenges typically arise when going beyond pairwise relationships. The experiment cover various topics but cover none in depth, which I think is the right call for this type of contributions. That being said, I think the paper would benefit from an empirical comparison between algorithms based on the proposed quantities and algorithms based on classical information quantities. At the moment the paper only compares run-time. Ideally, before using an alternative, faster algorithm, I'd like to have an idea how much the answer is going to differ. On a related note, it would be interesting for these experiments to illustrate the stability and robustness of the proposed quantities. In the synthetic data experiment with Gaussian vectors, what is the value of p and what is the family of density distributions used in the CL tree? Definition 2: Let e be an standard Gaussian random variable independent of Y --> Shouldn't e be capitalized? in equation (2), there is no "p" in the right hand side. Shouldn't it be p(y) in the second term? typos: l196 an standard l200 of the its counterpart ----------- I would like to thank the authors for their feedback. I think going beyond two variables would increase the interest of the work very much and I encourage them to include their additional results in the paper. I will keep my score unchanged.

Reviewer 3



ORIGINALITY: high QUALITY: high CLARITY: medium SIGNIFICANCE: medium The paper defines, analyzes, and applies new information theoretic quantities called gradient information. As we obtain Shannon entropy when we use the true distribution in cross entropy, we obtain gradient information entropy when we use the true distribution in the Hyvarinen formulation of Fisher divergence. Gradient information entropy is derived for common distributions (Table 1). We can further define conditional entropy and mutual information in similarly analogous ways and show that they together satisfy the chain rules. There are other nice properties: Gaussian is the maximal entropy distribution with a fixed variance (as in the classical case) and we can derive a Cramer-Rao bound-like result for mean squared error (Proposition 3). Because the Hyvarinen formulation bypasses normalization, we enjoy exactly the same computational benefits by using gradient information as in using the Hyvarinen loss. The authors demonstrate this by replacing the KL divergence with Fisher information in the Chow-Liu algorithm, yielding a substantial speed-up in runtime. Other applications on classification and community discovery are discussed. (An extension of the Hyvarinen formulation of Fisher divergence for bounded RVs is also presented, which is a nice complementary contribution.) One way the paper can improve is a more thorough discussion of 'why' these new quantities exhibit incongruences wrt the KL-based quantities. Is there a notion of cross entropy? Given how gradient information entropy is motivated, it may seem natural to define H_nabla(p,q) = E_{y~p}[s_nabla(y,q)]. Will it follow that H_nabla(Y) = min_q H_nabla(p,q)? I'm guessing that this is not included because it doesn't work out, but a discussion would be useful. Differential entropy is basically useless on its own (Marsh, 2013). Can we draw analogies with gradient information entropy, which isn't even positive? The stability interpretation is not sufficient for me to feel I know how to interpret these quantities. In the information bottleneck setting, learning p_{Z|Y} to maximize I(X,Z) where (x,y)~p_{XY} has a clear information theoretic meaning. Does gradient information mutual information have any application in this scenario? Due to such incongruences, we have yet to see if these new information theoretic measures will have significant impact in the future. Experiments suggest some promising directions. There are numerous minor errors in the manuscript which get in the way of reading. Please fix: they include - Change "log q(y)" to "log p(y)" in (2) - Change w to alpha in line 120. - In Definition 1, why say Y = [Y1, Y2]? - Include expectation when defining J(Y) in line 156. - What are the actual values of alpha and beta used in Table 1? - Section 3 can be written more clearly. UPDATE: Thanks for the response. Please do include either results or a discussion on cross entropy, which is by far the most practical quantity in the classical case.

[Author Response · NeurIPS 2019]

## Response to Reviewer 1

We sincerely thank Reviewer 1 for referring us to four relevant papers [1-4]. We will include a detailed discussion on these papers in our revised manuscript, but would like to emphasize that our work differs from, and is complementary to this literature. This is further explained below:

Paper [1] provides a very interesting relationship between Fisher divergence and Stein's operator, whereby Hyvarinen score and its extensions are cast as specific Stein's operators. Paper [4] establishes a more general result than that of S. Lyu.

In contrast, our work focuses on utilizing Hyvarinen score (and its extensions) to develop information quantities for both interpretability and computational benefits (e.g. in our example of fast tree approximation). That said the computational motivation still remains to be the getting rid of normalizing constants.

Regarding your first technical question: It follows from an application of weak law of large number that by minimizing the Hyvranin score the parametric density converges to the shadow of the true data generating distribution under mild assumptions. If the model class is well-specified then convergence to the data generating distribution is guaranteed.

Regarding your second technical question raised: The Fisher entropy is intimately related to the Cramer-Rao Bound in parameter estimation. Fano's inequality also gives lower bounds of model selection/message decoding error (so larger bound implies more complexity for 'description'). We understand that Reviewer 1 may have a different interpretation of our wording, and we will clarify this in the revision.

[1] Liu, Qiang, Jason Lee, and Michael Jordan. 'A kernelized Stein discrepancy for goodness-of-fit tests.' ICML 2016.

[2] Chwialkowski, Kacper, Heiko Strathmann, and Arthur Gretton. 'A kernel test of goodness of fit.' ICML 2016.

[3] Gorham, Jackson, and Lester Mackey. 'Measuring sample quality with kernels.' ICML 2017.

[4] Liu, Qiang, and Dilin Wang. 'Stein variational gradient descent: A general purpose bayesian inference algorithm.' NIPS 2016.

## Response to Reviewer 2

We appreciate Reviewer 2's comments and recommendations.

We will do another proof reading and remove typos. Due to page limit we could not include more extensive experiments, but we plan to do so in subsequent works.

Reviewer 2 raised an excellent point about the extension from pairs of variables to groups of variables. We have some recent results on this (not reported due to page limit), and hope to expand on them and present them in future work.

Regarding the two questions raised by the Reviewer 2, the values of p ranges from 2 to 11 and the joint density is determined by the conditional exponential family distribution in (12) and the discovered tree structure; the second term in (2) is q(y), and p(y) has been implicitly included in the expectation.

## Response to Reviewer 3

We appreciate Reviewer 3's comments and recommendations.

Reviewer 3 raised an excellent point about defining cross entropy. We think it is doable and it would be another analogy to the classical cross entropy. Despite theoretical interest, we have not yet found a real data application for this. We will include a discussion of this in the revision.

Like the point made in (Marsh 2013), it is also true in our case that the relative entropy between distributions is more interpretable than entropy on its own. We have not found an application of gradient mutual information in the settings of information bottleneck. But we believe this direction is extremely interesting. As suggested by Reviewer 3, we will do another proof reading and remove the typos.

[Meta-Review · NeurIPS 2019]

The reviewers and I agree on the contribution of this paper. However, there are still some concerns and suggestions that would extremely benefit the quality of this submission. The authors are strongly encouraged to address these points before the camera ready paper: - Please add more discussions on existing works on the Stein score function. - Please aim to go beyond two variables - Please do include results or a discussion on cross entropy